



**Aerosol transport pathways and source attribution in China**
**during the COVID-19 outbreak**
Lili Ren[1], Yang Yang[1*], Hailong Wang[2], Pinya Wang[1], Lei Chen[1], Jia
Zhu[1], Hong Liao[1]
[1]Jiangsu Key Laboratory of Atmospheric Environment Monitoring and Pollution
Control, Jiangsu Collaborative Innovation Center of Atmospheric Environment and
Equipment Technology, School of Environmental Science and Engineering, Nanjing
University of Information Science and Technology, Nanjing, Jiangsu, China
[2]Atmospheric Sciences and Global Change Division, Pacific Northwest National
Laboratory, Richland, Washington, USA

21 *Correspondence to yang.yang@nuist.edu.cn



**Abstract**

Due to the coronavirus disease 2019 (COVID-19) pandemic, human activities and industrial productions were strictly restricted during January-March 2020 in China. Despite the fact that anthropogenic aerosol emissions largely decreased, haze events still occurred. Characterization of aerosol transport pathways and attribution of aerosol sources from specific regions are beneficial to the air quality and pandemic control strategies. This study establishes source-receptor relationships in various regions of China during the COVID-19 outbreak based on the Community Atmosphere Model version 5 with Explicit Aerosol Source Tagging (CAM5-EAST). Our analysis shows that $PM_{2.5}$ burden over the North China Plain between January 30 and February 19 is largely contributed by local emissions (40–66%). For other regions in China, $PM_{2.5}$ burden is largely contributed from non-local sources. During the polluted days of COVID-19 outbreak, local emissions within North China Plain and Eastern China, respectively, contribute 66% and 87% to the increase in surface $PM_{2.5}$ concentrations. This is associated with the anomalous mid-tropospheric high pressure at the location of climatological East Asia trough and the consequently weakened winds in the lower troposphere, leading to the local aerosol accumulation. The emissions outside China, especially from South and Southeast Asia, contribute over 50% to the increase in $PM_{2.5}$ concentration in Southwestern China through





transboundary transport during the polluted day. As the reduction in
emissions in the near future, aerosols from long-range transport together
with unfavorable meteorological conditions are increasingly important to
regional air quality and need to be taken into account in clean air plans.





## 1. Introduction


The coronavirus disease 2019 (COVID-19) had an outbreak in China
in December 2019. It has resulted in more than one million cases within
the first four months worldwide (Sharma et al., 2020; Dong et al., 2020).
In order to curb the virus spread among humans, China was the first
country to take dramatic measures to minimize the interaction among
people, including strict isolation, prohibition of large-scale private and
public gatherings, restriction of private and public transportation and even
lockdown of cities (Tian et al., 2020; Wang et al., 2020). The estimated
NOx emission in eastern China was reduced by 60-70%, of which 70-80%
was related to the reduced road traffic and 20-25% was from industrial
enterprises shutdown during the COVID-19 lockdown period (Huang et al.,
2020). However, severe air pollution events still occurred in East China
during the COVID-19 lockdown. It is of great concern that why severe air
pollution was not avoided by decreasing anthropogenic emissions.
The unprecedented large-scale restrictions resulting from the COVID-
19 epidemic provide an opportunity to research the relationship between
dramatic anthropogenic emission reductions and air quality change (e.g.,
Bao et al., 2020; Li et al., 2020; Wang et al., 2020). Bao et al. (2020)
reported that, during the COVID-19 lockdown period, the air quality index
(AQI) and the $PM_{2.5}$ (particulate matter less than 2.5 μm in diameter)
concentration were decreased by 7.8% and 5.9 %, respectively, on average



in 44 cities in northern China, mainly due to travel restrictions. By applying
the WRF-CAMx model together with air quality monitoring data, Li et al.
(2020) revealed that although primary particle emissions were reduced by
15%–61% during the COVID-19 lockdown over the Yangtze River Delta
Region, the daily mean concentration of $PM_{2.5}$ was still relatively high,
reaching up to 79 μg m$^{-3}$. Wang et al. (2020) found that the relative
reduction in $PM_{2.5}$ precursors was twice as much as the reduction in $PM_{2.5}$
concentration, in part due to the unfavorable meteorological conditions
during the COVID-19 outbreak in China that led to the formation of the
heavy haze. Huang et al. (2020) and Le et al. (2020) reported that stagnant
air conditions, high atmospheric humidity, and enhanced atmospheric
oxidizing capacity led to a severe haze event in northern China during the
COVID-19 pandemic.
Aerosols are main air pollutants that play important roles in the
atmosphere due to their adverse effects on air quality, visibility (Vautard et
al., 2009; Watson, 2002), human health (Lelieveld et al., 2019; Heft-Neal
et al., 2018), the Earth's energy balance, and regional and global climate
(Ramanathan et al., 2001; Anderson et al., 2003; Wang et al., 2020; Smith
et al., 2020). With the rapid development in recent decades, China has
experienced severe air pollutions that damage human health and cause
regional climate change (Chai et al., 2014; Liao et al., 2015; Fan et al.,
2020). In order to control air pollution, the Chinese government issued and



implemented the Air Pollution Prevention and Control Action Plan in 2013
(China State Council, 2013). Although emissions in China have decreased
significantly in recent years (Zheng et al., 2018), aerosols transported from
other source regions could add on top of local emissions (Yang et al., 2017a,
2018a; Ren et al., 2020). Therefore, it is important to understand the
relative effects of local emissions and regional transport on aerosols in
China.
Source tagging and apportionment is an effective way to establish
aerosol source-receptor relationships, which is conducive to both scientific
research and emission control strategies (Yu et al., 2012). By applying the
Particulate Source Apportionment Technology in CAMx model, Xue et al.
(2014) found that the contributions of regional transport to annual average
$PM_{2.5}$ concentrations in Hainan, Shanghai, Jiangsu, Zhejiang, Jilin and
Jiangxi provinces of China are more than 45%. By adding a chemical tracer
into the WRF model, Wang et al. (2016) studied the sources of black carbon
(BC) aerosol in Beijing and reported that about half of BC in Beijing came
from the central North China Plain. Liu et al. (2017) applied WRF-Chem
model and showed that Foshan, Guangzhou and Dongguan, respectively,
with relatively high emissions contributed 14%, 13% and 10% to the
regional mean $PM_{2.5}$ concentration in the Pearl River Delta.
Currently, many studies have investigated the impact of reduced human
activity on regional air quality, as a result of the COVID-19 outbreak. Few



studies have focused on aerosol transport pathways and source attribution
in China during the COVID-19 pandemic. In this study, the global aerosol-
climate model CAM5 (Community Atmosphere Model, version 5)
equipped with an Explicit Aerosol Source Tagging (CAM5-EAST) is
employed to quantify source-receptor relationships and transport pathways
of aerosols during the COVID-19 outbreak in China. We also provide
model evaluations of $PM_{2.5}$ concentrations against observations made
during the COVID-19 outbreak. With the aerosol source tagging technique,
source region contributions to $PM_{2.5}$ column burden over various receptor
regions and transport pathways in China are analyzed. The source
contributions to the changes in near-surface $PM_{2.5}$ in polluted days
compared to the monthly means during February 2020 are also quantified.
This paper provides source apportionment of aerosols in China during the
COVID-19 emission reductions, which is beneficial to the investigation of
policy implications for future air pollution control.
**2. Methods**
**2.1 Model description and experimental setup**

The CAM5 model is applied to estimate the $PM_{2.5}$ changes during the

COVID-19 period. In CAM5, which is the atmospheric component of the
earth system model CESM (Community Earth System Model, Hurrell et
al., 2013). In this study, major aerosol species including sulfate, BC,
primary organic matter (POM), secondary organic aerosol (SOA), sea salt,



and mineral dust, are represented by three lognormal size modes (i.e.,
Aitken, accumulation, and coarse modes) of the modal aerosol module
(MAM3) (Liu et al., 2012). The detailed aerosol representation in CAM5
was provided in Liu et al. (2012) and Wang et al. (2013). The aerosol
mixing states consider both internal mixed (within a same mode) and
external mixed (between modes). On top of the default CAM5, additional
modifications that improve the representation of aerosol wet scavenging
and convective transport (Wang et al., 2013) are also included in the model
version used for this study.
In this study, simulations were conducted with a horizontal resolution
of 1.9° × 2.5° and 30 vertical layers up to 3.6 hPa in year 2020.
Anthropogenic emissions in China are derived from the MEIC (Multi-
resolution Emission Inventory of China) inventory (Zheng et al., 2018).
while emissions for the other countries use the SSP (Shared Socioeconomic
Pathways) 2–4.5 scenario data set under CMIP6 (the Coupled Model
Intercomparison Project Phase 6). Emissions in year 2017 are used as the
baseline during the simulation period considering the time limit of MEIC
inventory. To better estimate the impact of restricted human activities on
emission reductions owing to COVID-19 lockdown, we updated China's
emission inventory from January to March 2020 based on the provincial
total emission reduction ratio in Huang et al. (2020). Emissions from the
transportation sector are decreased by 70% and the remaining reductions



159 are evenly distributed to other sectors from January to March 2020

160 compared to the baseline emission in 2017. The sea surface temperature,

161 sea ice concentrations, solar radiation and greenhouse gas concentrations

162 are fixed at present-day climatological levels. To capture the large-scale

163 atmospheric circulations during the COVID-19, we nudge the model wind

164 fields toward the MERRA-2 (Modern-Era Retrospective Analysis for

165 Research and Applications, version 2) reanalysis (Gelaro et al., 2017) from

166 April 2019 to March 2020 repeatedly for six years. Only model results from

167 the last year are used to represent year 2020. In this study, we analyze the

168 transport pathways and source attribution of aerosols during the three

169 weeks that had the largest number of newly-diagnosed COVID-19 cases

170 (Fig. 2, hereafter referred to as the 'Week 1': January 30–February 5,

171 'Week 2': February 6–February 12 and 'Week 3': February 13–February

172 19), when unexpected hazardous air pollution events also occurred during

173 this time period (Le et al., 2020).

**2.2 Explicit aerosol source tagging and source regions**

175 To examine the source apportionment of aerosols in China, the Explicit

176 Aerosol Source Tagging (EAST) technique was implemented in CAM5,

177 which has been utilized in many aerosol source attribution studies (e.g.,

178 Wang et al., 2014; Yang et al., 2017a, b, 2018a, b, c, 2020; Ren et al., 2020).

179 Different from the emission sensitivity method that assumes a linear

180 response to emission perturbation and the traditional backward trajectory




method, aerosols from each tagged region or sector are calculated
independently in EAST within one single simulation. Without relying on a
set of model simulations with emission perturbations or assuming constant
decaying rate, EAST is more accurate and time-saving than the source
apportionment method mentioned above. In addition to the sulfate, BC and
POM species that were tagged in previous studies (e.g., Yang et al., 2020),
SOA and precursor gas are now also tagged in the EAST. These types of
aerosols from independent source regions and sectors can be explicitly
tagged and tracked simultaneously. In this study, focusing on the aerosols
in China during the COVID-19 outbreak period, the domestic aerosol and
precursor emissions are divided into eight geographical source regions (Fig.
1), including Northeastern China (NEC), North China Plain (NCP), Eastern
China (ESC), Southern China (STC), Central-West China (CWC),
Southwestern China (SWC), Northwestern China (NWC) and the
Himalayas and Tibetan Plateau (HTP), and the rest of the world (ROW)
emissions are tagged separately.
**3. Model evaluation**
Many previous studies have assessed the spatial distribution and
seasonal to decadal variations in aerosol concentrations in China and
worldwide simulated by CAM5 with the observations (e.g., Wang et al.,
2013; Yang et al., 2017a,b, 2018b,c, 2020). In order to evaluate the model's
performance in simulating aerosols during the COVID-19 outbreak period



in China, the surface concentrations of $PM_{2.5}$, estimated as the sum of
sulfate, BC, POM and SOA for model results, during the analyzed time
periods are compared with measurements from the China National
Environmental Monitoring Center (CNEMC), as shown in Fig. 3a. The
model reasonably reproduces the overall spatial distribution of near-
surface $PM_{2.5}$ concentrations during the three time periods, with high
values in the North China Plain and low values in western China. However,
as reported in many CAM5 model studies (e.g., Yang et al., 2017a,b), the
model underestimates the $PM_{2.5}$ concentrations with normalized mean
biases (NMB) of -55%~-49%, compared to the available site observations
(Fig. S1). The discrepancies are related to coarse-resolution model
sampling bias relative to the observational sites, uncertainties in aerosol
emissions, wet removal, and gas-particle exchange. In addition, the model
version used in this study is not able to simulate nitrate and ammonium
aerosols, which are also the main components of $PM_{2.5}$ (Kong et al., 2020;
Xu et al., 2019).
The long-distance transport of aerosols mainly occurs in the upper
troposphere rather than near the surface (Hadley et al., 2007; Zhang et al.,
2015). Aerosols are lifted from the atmospheric boundary layer of the
emission source regions to the free troposphere and then undergo the
transboundary and intercontinental transport effectively driven by the
upper tropospheric circulations. Therefore, it is helpful to analyze the



relative contributions of local and non-local sources by focusing on the
column burden of aerosols. Figure 3b presents spatial distributions of
simulated mean column burden of $PM_{2.5}$ during the three time periods. The
contrast in column burden does not differ significantly from that of near-
surface concentrations. Among the three time periods, Week 1 and Week 2
have higher $PM_{2.5}$ loading, with values in the range of 20–40 and 20–30
mg m$^{-2}$ in the North China Plain, Eastern China, and Southern China, while
the $PM_{2.5}$ loading in Week 3 is relative lower with values ranging mostly
from 10 to 20 mg m$^{-2}$. Note that the column burden of $PM_{2.5}$ in South and
Southeast Asia is higher than 20 mg m$^{-2}$ in three time periods and reaches
up to 50 mg m$^{-2}$ in Week 2, which potentially influences aerosol
concentrations in China through transboundary transport.
**4. Transport Pathways**
The explicit aerosol tagging technique can clearly identify the transport
pathways of aerosols moving from their source regions to their destination.
Figure 4 shows the spatial distribution of mean column burden of simulated
$PM_{2.5}$ originating from the six tagged source regions in central and eastern
China and outside of China during the three time periods. Aerosols and/or
precursor gases emitted from the various regions follow quite different
transport pathways determined by their source locations, meteorological
conditions, emission injection height, and physical and chemical
characteristics of aerosol species. Aerosols from Northeastern China are



transported southeastward by the northwesterly winds (Fig. 1b). From the
North China Plain, aerosols can be transported either southward reaching
Eastern, Southern and Southwestern China during Week 1 or across east
coast of China to the oceanic region during Week 2-3. Aerosols originating
from Eastern China move straight to Southwestern and Southern China
during Week 1-2, while they also entered the North China Plain during
Week 2-3. Aerosols emitted from Southern China and Central-West China
have no obvious transport due to their relatively weak emissions. In
additional to the local impact, emissions from Southwestern China affect
mostly the Southern China and Eastern China. Air parcels with high levels
of $PM_{2.5}$ from South and Southeast Asia moved into Southwestern,
Southern and Eastern China and even the North China Plain during the
three time periods.

The vertical distributions of $PM_{2.5}$ emitted from six major tagged

source regions are shown in Figs. S2 and S3. The $PM_{2.5}$ has much higher
concentrations in the lower troposphere and decreases with increasing
height. During Week 1-2, owing to the presence of high $PM_{2.5}$ loadings, a
stronger vertical mixing and transport brought more $PM_{2.5}$ to the upper
troposphere compared to that during Week 3. High concentrations of $PM_{2.5}$
originating from the North China Plain extended southeastward by strong
northwesterly winds. Weak winds over Eastern China led to accumulations
of $PM_{2.5}$ within this region, which is consistent with the findings in Yang



et al. (2017a). Strong southwesterly winds in the south of Southwestern
China and weak winds in the north of this region produced convergences
and updrafts that lift aerosols up to 700 hPa.
Considering that the emissions outside China contribute greatly to
$PM_{2.5}$ concentrations in Southwestern China through transboundary
transport (Yang et al., 2017a) and aerosols from East Asia can be
transported to the North Pacific and even North America (Yu et al., 2008;
Yang et al., 2018c), it is of great importance to study the inflow and outflow
of $PM_{2.5}$ across the boundaries of China. Figures 5 and 6 show the vertical
distribution of $PM_{2.5}$ concentrations resulting from emissions within and
outside China over 29°N,88°E and 21°N around the south boundaries
(cross-sections (CS) 1-3 in Fig. 1a) and 123° E around the east boundary
(CS 4 in Fig. 1a) of the mainland of China. Over the southern border, $PM_{2.5}$
concentrations are more influenced by transboundary transport of aerosols
from ROW than those originating from domestic emissions. The high
concentrations of $PM_{2.5}$ from South and Southeast Asia are lifted into the
free atmosphere of the Tibetan Plateau and Yun-Gui Plateau, and then
transported to Southern and Southwestern China by southwesterly winds.
Over the North China Plain and Eastern China, northwesterly winds at 35-
45° N and southwesterly winds at 25-35° N cause aerosols to accumulate
in the lower atmosphere and then export across east border of China below
700 hPa.



## 5. Source apportionment of PM$_{2.5}$ in China during the COVID-19

### 5.1 Source contributions to PM$_{2.5}$ burden

Figure 7 shows the simulated relative contributions in percentage to PM$_{2.5}$ column burden from local source emissions, regional transport from the untagged regions of China (rest of China, RCN) and rest of the world (ROW). Over the North China Plain, where emissions are relatively high, PM$_{2.5}$ column burden is dominated by local emissions during the three time periods. In contrast, regions with relative low emissions are mainly affected by nonlocal sources, especially by foreign contributions. Emissions from the ROW contribute a large amount to PM$_{2.5}$ burden over Northeastern, Southern, Central-West, Southwestern, Northwestern China and the Tibetan Plateau. PM$_{2.5}$ burden in Eastern China is greatly contributed by the sources from RCN, especially in Week 1 when regional transport of PM$_{2.5}$ from the North China Plain is relatively strong (Fig. S3).

Table 1 summarizes the contributions of tagged source regions to the PM$_{2.5}$ burden over different receptor regions in China. In Northeastern China, 36%-43% of the PM$_{2.5}$ column burden comes from local emissions, while a larger portion (39%-54%) is contributed by emissions from ROW during the three time periods. The impacts of nonlocal sources within China on PM$_{2.5}$ burden are relatively low in Northeastern China during Week 1 with the contribution of less than 5%, while RCN is responsible for 23% and 25% during Week 2 and Week 3, respectively.



In the North China Plain, the majority of the PM$_{2.5}$ burden is attributed
to local emissions in all cases, with local contributions in a range of 40–
66%. Emissions from the North China Plain also produce a widespread
impact on PM$_{2.5}$ over its neighboring regions. The sources from North
China Plain account for 14–33% of the PM$_{2.5}$ burden in Eastern China and
7–23% in Southern China during the three time periods.
In Eastern China, local emissions account for 27–40% of PM$_{2.5}$ column
burden, while ROW contributes 20–45%. Southern China and Central-
West China have 13–18% and 25–31% of local source contributions,
respectively, whereas 37–64% are due to emissions from outside China in
these two regions. In Southwestern China, 15–18% of the PM$_{2.5}$ burden
originates from local emissions and 7–24% is from RCN. ROW emissions
play important roles in affecting PM$_{2.5}$ burden over this region, with
relative contributions in a range of 59–78% during the three time periods,
which is associated with the transboundary transport by southwesterly
winds. PM$_{2.5}$ burden over the Northwestern China and Himalayas and
Tibetan Plateau with relatively low local emissions are strongly influenced
by nonlocal sources, where more than 70% of the PM$_{2.5}$ burden originates
from emissions outside China.
**5.2 Aerosol source attribution during polluted days**
In spite of the large reductions in emissions, severe air pollution events
still occurred in China during the COVID-19 lockdown. Source attribution





of PM$_{2.5}$ during polluted days in China has policy implications for future
air pollution control. In Beijing, capital of China over the North China
Plain, a serious haze event happened from February 11 to 13, 2020 during
the COVID-19 outbreak period according to observations released by
CNEMC. CAM5-EAST reproduced the polluted day on February 11 over
the North China Plain. In this study, the most polluted day is defined as the
day with the highest daily PM$_{2.5}$ concentration in February 2020 for each
receptor region in China. Figure 8 presents the composite differences in
near-surface PM$_{2.5}$ concentrations and 850 hPa wind fields between
polluted days and normal days (all days in February 2020) for each receptor
region. The local and nonlocal source contributions to the PM$_{2.5}$ differences
are summarized in Fig. 9.
Unexpectedly, near-surface PM$_{2.5}$ concentrations in the North China
Plain and Eastern China experienced remarkable increases during the
polluted days of COVID-19 lockdown. The simulated PM$_{2.5}$ concentrations
increased, with the largest increases of more than 20 μg m$^{-3}$ in the North
China Plain and Eastern China, 10 μg m$^{-3}$ maximum increase in the
Southwestern China and 5 μg m$^{-3}$ in the Northeastern, Southern and
Central-West China, during the most polluted days compared to the normal
days.
The increase in near-surface PM$_{2.5}$ concentrations during the most
polluted day over Northeastern China is largely influenced by the local



emissions, which contribute to a regional averaged concentration increase
of 1.1 μg m$^{-3}$. This is mainly due to the accumulation of local aerosols
under the weakened prevailing northwesterly winds over this region.
When the PM$_{2.5}$ pollution occurred in the North China Plain, the
concentration of PM$_{2.5}$ was 16.1 μg m$^{-3}$ higher than that in normal days.
The contribution from local emissions accounts for 66% of the averaged
increase, which was related to the stagnant air condition (i.e., weakened
lower tropospheric winds) resulting from the anomalous mid-tropospheric
high pressure located at the climatological location of the East Asia trough
(Fig. S4). Sources from Eastern China also explain 4.3 μg m$^{-3}$ (27%) of the
total increase over the North China Plain.
During the most polluted day in Eastern China (the same day as the
polluted day in North China Plain), the regional averaged increase in PM$_{2.5}$
concentrations is 16 μg m$^{-3}$, which is primarily contributed by the local
emissions. While the contribution from the North China Plain decreased in
the polluted day, the anomalous southerly winds brought more aerosols
from Southern China and ROW into Eastern China, contributing to 4% and
10% aerosol increase, respectively.
Owing to the enhanced northerly winds, emissions from the North
China Plain and Eastern China contribute 33% and 39% of the increase,
respectively, in PM$_{2.5}$ concentration over Southern China. The most
polluted day in Central-West China is mostly caused by local emissions


(65% of the total increase).
When Southwestern China was under the polluted condition, $PM_{2.5}$
concentration was increased by 2.1 μg m$^{-3}$. Emissions from ROW,
especially those from South and Southeast Asia, are of great significance
to the increase of $PM_{2.5}$ concentrations due to the enhanced southwesterly
winds over this region. The relative contribution from ROW emissions is
more than 50% over Southwestern China during the most polluted day. It
highlights that the important role of transboundary transport needs to be
considered when controlling local emissions to improve air quality in the
near future.

**6. Conclusions and discussions**

An explicit aerosol source tagging is implemented in the Community
Atmosphere Model version 5 (CAM5-EAST) to examine the aerosol
transport pathways and source attribution of $PM_{2.5}$ in China during the first
few weeks of the COVID-19 outbreak (Week 1: January 30–February 5,
Week 2: February 6–February 12 and Week 3: February 13–February 19).
The contributions of emissions to $PM_{2.5}$ originating from eight source
regions in the mainland of China, including Northeastern China, North
China Plain, Eastern China, Southern China, Central-West China,
Southwestern China, Northwestern China and Himalayas and Tibetan
Plateau, and sources outside China (ROW) to near-surface concentrations,



column burdens, transport pathways of $PM_{2.5}$, and haze formation in
different receptor regions in China are quantified in this study.

Aerosols emitted from the North China Plain, where the air quality is

often poor, can be transported through Eastern China and reach
Southwestern China during the three time periods. Similarly, aerosols from
Eastern China move straight to Southern China and Southwestern China
during Week 1 and Week 2, and a significant portion can also enter the
North China Plain during Week 2 and Week 3.

Across the southern boundary of the mainland of China, high

concentrations of $PM_{2.5}$ from South and Southeast Asia are lifted into the
free atmosphere and then transported to Southern and Southwestern China.
While $PM_{2.5}$ from the North China Plain and Eastern China can also be
brought out of China via westerly winds, mostly below 700 hPa.

$PM_{2.5}$ in China is affected not only by local emissions but also by long-

range transport of pollutants from distant source regions. Over the North
China Plain, 40–66% of the $PM_{2.5}$ burden is attributed to local emissions
during the COVID-19 outbreak. They also impact $PM_{2.5}$ in neighboring
regions, accounting for 14–33% of the $PM_{2.5}$ burden in Eastern China and
7–23% in Southern China during the three time periods. Northeastern
China has 36%-43% of local source contributions to its $PM_{2.5}$ column
burden, while 39%-54% is contributed by emissions from ROW during the
three time periods. In Eastern China, local emissions explain 27–40% of



$PM_{2.5}$ burden, while ROW contributes 20–45%. In Southwestern China,
59–78% of the $PM_{2.5}$ burden is contributed by emissions from ROW. Over
the Northwestern China and Himalayas and Tibetan Plateau, ROW
emissions have a great contribution of more than 70% to the $PM_{2.5}$ column
burden.
Despite the large reductions in emissions, near-surface $PM_{2.5}$
concentrations in the North China Plain and Eastern China increased a lot
during the most polluted days of COVID-19 lockdown (with the highest
daily $PM_{2.5}$ concentration in February 2020), with the largest increases of
more than 20 μg m$^{-3}$. In addition to local emissions, regional transport of
pollutants is also an important factor that causes haze events in China. The
increases in $PM_{2.5}$ concentrations during the most polluted days over the
North China Plain and Eastern China are largely influenced by the stagnant
air condition resulting from the anomalous high pressure system and
weakening of winds, which lead to a reduced ventilation and aerosol
accumulation in the North China Plain, together with an increase in aerosol
inflow from regional transport. During the most polluted day in
Southwestern China, ROW contributes over 50% of the $PM_{2.5}$
concentration increase, with enhanced southwesterly winds that drive
pollution transport from South and Southeast Asia. It indicates that regional
transport and unfavorable meteorology need to be taken into consideration
when controlling local emissions to improve air quality in the near future.



There are a few uncertainties in this study. The CAM5 model has low
biases in reproducing the near-surface $PM_{2.5}$ concentrations in China,
compared to observations, in part due to the incapability of simulating
some aerosol components of $PM_{2.5}$ (e.g., ammonium and nitrate), excessive
aerosol wet removal during the long-range transport (Wang et al., 2013),
and uncertainties in observations. Uncertainties in the estimate of emission
reductions in different source regions during the COVID-19 pandemic can
also introduce uncertainties to our results. During the COVID-19 lockdown,
greenhouse gas emissions also decreased (Le Quéré et al., 2020), but the
effect of greenhouse gas reduction on meteorology that potentially
influence aerosol distributions was not taken into consideration.
Nevertheless, this study is the first attempt to provide source
apportionment of aerosols in China during the COVID-19 outbreak, which
is beneficial to the investigation of policy implications for future air
pollution control.
***Data availability.***
The CAM5 model is available at
http://www.cesm.ucar.edu/models/cesm1.2/ (last access: 25 October 2020).
CAM5-EAST model code and results can be made available upon request.
The surface $PM_{2.5}$ observations are from the China National Environmental
Monitoring Center (CNEMC, http://www.cnemc.cn, last access: 25
October 2020)
***Competing interests.***
The authors declare that they have no conflict of interest.
***Author contribution***.
YY and LR designed the research; YY performed the model simulations;
LR analyzed the data. All authors discussed the results and wrote the paper.
***Acknowledgments.***
This study was supported by the National Key Research and Development
Program of China (grant 2020YFA0607803) and the National Natural
Science Foundation of China (grant 41975159). HW acknowledges the
support by the U.S. Department of Energy (DOE), Office of Science,
Office of Biological and Environmental Research (BER). The Pacific
Northwest National Laboratory (PNNL) is operated for DOE by the
Battelle Memorial Institute under contract DE-AC05-76RLO1830.



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





**Table 1.** Fractional contributions of emissions from nine tagged source regions (vertical
axis) to mean PM$_{2.5}$ column burden in eight receptor regions (horizontal axis) during
the three time periods.

**Week 1**

| source | NEC | NCP | ESC | STC | CWC | SWC | NWC | HTP |
|---|---|---|---|---|---|---|---|---|
| NEC | 0.43 | 0.02 | 0.01 | 0.01 | 0.00 | 0.00 | 0.00 | 0.00 |
| NCP | 0.01 | 0.66 | 0.33 | 0.23 | 0.07 | 0.09 | 0.00 | 0.00 |
| ESC | 0.00 | 0.04 | 0.38 | 0.15 | 0.05 | 0.09 | 0.00 | 0.00 |
| STC | 0.00 | 0.00 | 0.01 | 0.18 | 0.00 | 0.03 | 0.00 | 0.00 |
| CWC | 0.00 | 0.05 | 0.02 | 0.01 | 0.27 | 0.02 | 0.00 | 0.00 |
| SWC | 0.01 | 0.01 | 0.04 | 0.05 | 0.12 | 0.17 | 0.01 | 0.01 |
| NWC | 0.01 | 0.01 | 0.00 | 0.00 | 0.04 | 0.00 | 0.19 | 0.01 |
| HTP | 0.00 | 0.01 | 0.00 | 0.00 | 0.04 | 0.00 | 0.03 | 0.03 |
| ROW | 0.54 | 0.20 | 0.20 | 0.37 | 0.41 | 0.59 | 0.77 | 0.95 |

**Week 2**

| source | NEC | NCP | ESC | STC | CWC | SWC | NWC | HTP |
|---|---|---|---|---|---|---|---|---|
| NEC | 0.36 | 0.01 | 0.00 | 0.00 | 0.00 | 0.00 | 0.00 | 0.00 |
| NCP | 0.18 | 0.59 | 0.17 | 0.07 | 0.06 | 0.02 | 0.00 | 0.00 |
| ESC | 0.02 | 0.13 | 0.40 | 0.13 | 0.07 | 0.08 | 0.00 | 0.00 |
| STC | 0.00 | 0.00 | 0.03 | 0.17 | 0.00 | 0.04 | 0.00 | 0.00 |
| CWC | 0.01 | 0.07 | 0.02 | 0.01 | 0.25 | 0.00 | 0.00 | 0.00 |
| SWC | 0.00 | 0.02 | 0.06 | 0.07 | 0.09 | 0.18 | 0.00 | 0.01 |
| NWC | 0.01 | 0.01 | 0.00 | 0.00 | 0.03 | 0.00 | 0.17 | 0.01 |
| HTP | 0.00 | 0.01 | 0.00 | 0.00 | 0.04 | 0.00 | 0.03 | 0.03 |
| ROW | 0.41 | 0.17 | 0.31 | 0.55 | 0.46 | 0.68 | 0.80 | 0.95 |

**Week 3**

| source | NEC | NCP | ESC | STC | CWC | SWC | NWC | HTP |
|---|---|---|---|---|---|---|---|---|
| NEC | 0.36 | 0.02 | 0.00 | 0.00 | 0.00 | 0.00 | 0.00 | 0.00 |
| NCP | 0.15 | 0.40 | 0.14 | 0.09 | 0.01 | 0.02 | 0.00 | 0.00 |
| ESC | 0.06 | 0.19 | 0.27 | 0.10 | 0.02 | 0.03 | 0.00 | 0.00 |
| STC | 0.00 | 0.02 | 0.07 | 0.13 | 0.01 | 0.01 | 0.00 | 0.00 |
| CWC | 0.02 | 0.04 | 0.01 | 0.00 | 0.31 | 0.01 | 0.00 | 0.00 |
| SWC | 0.02 | 0.06 | 0.06 | 0.03 | 0.11 | 0.15 | 0.01 | 0.01 |
| NWC | 0.01 | 0.01 | 0.00 | 0.00 | 0.04 | 0.00 | 0.16 | 0.01 |
| HTP | 0.00 | 0.00 | 0.00 | 0.00 | 0.00 | 0.00 | 0.02 | 0.04 |
| ROW | 0.39 | 0.25 | 0.45 | 0.64 | 0.45 | 0.78 | 0.80 | 0.93 |

receptor


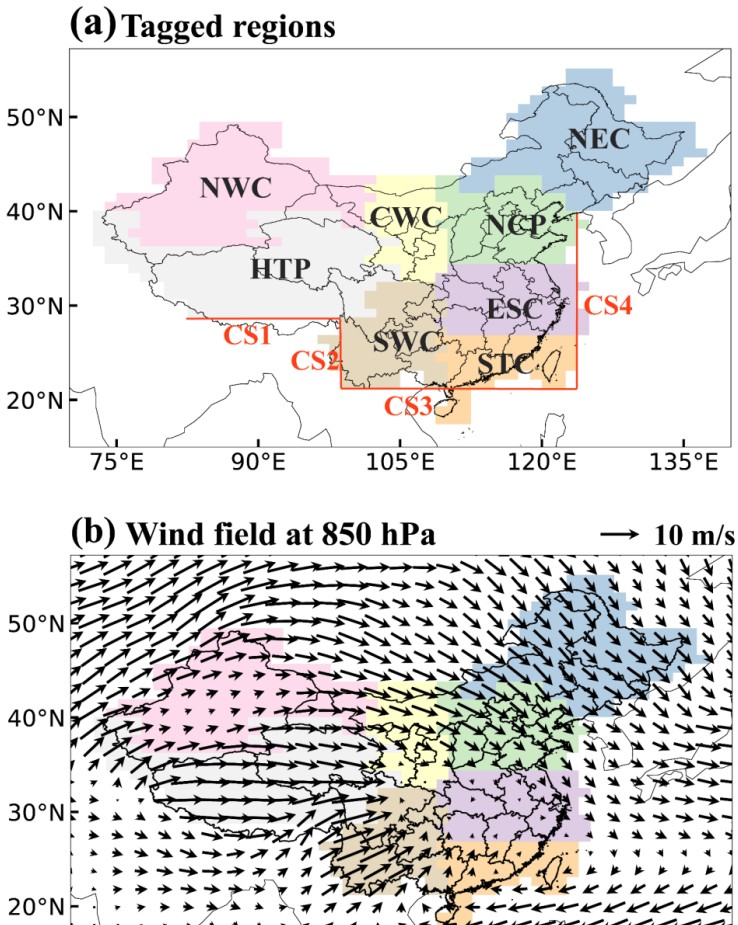

**Figure 1.** (a) Tagged source regions (NEC: Northeastern China, NCP: North China Plain, ESC: Eastern China, STC: Southern China, CWC: Central-West China, SWC: Southwestern China, NWC: Northwestern China, HTP: Himalayas and Tibetan Plateau, ROW: rest of the world) and (b) mean wind field (units: m s$^{-1}$, vectors) at 850 hPa during the time period of interest. Lines in (a) mark the cross-sections (CS) defined to study the transport of aerosols to and from China.



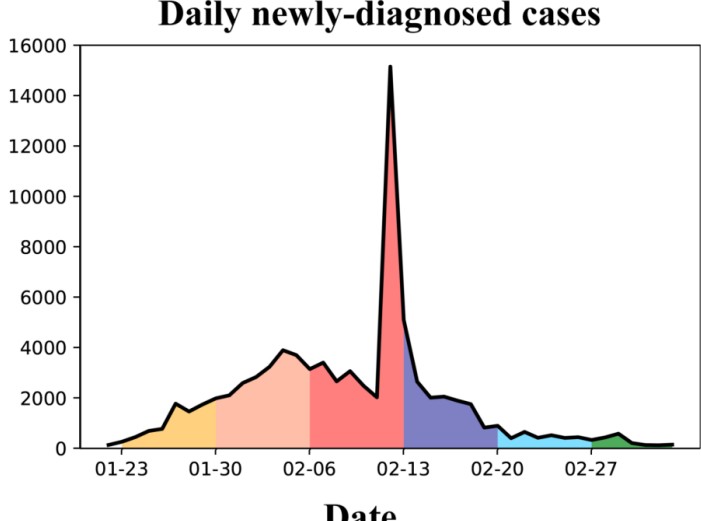



**Figure 2.** The number of daily newly-diagnosed cases in China from January 23 to
February 27, 2020, during the COVID-19.

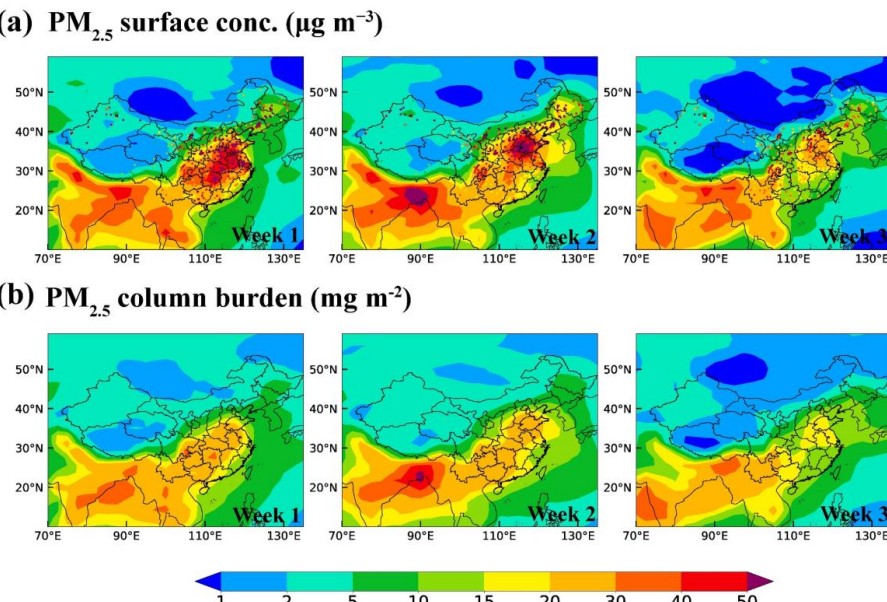

**Figure 3.** Spatial distribution of (a) the simulated and observed mean near-surface
$PM_{2.5}$ concentrations ($\mu g\ m^{-3}$) and (b) $PM_{2.5}$ column burden ($mg\ m^{-2}$) during January
30–February 5 (Week 1), February 6–February 12 (Week 2) and February 13–February
19 (Week 3).



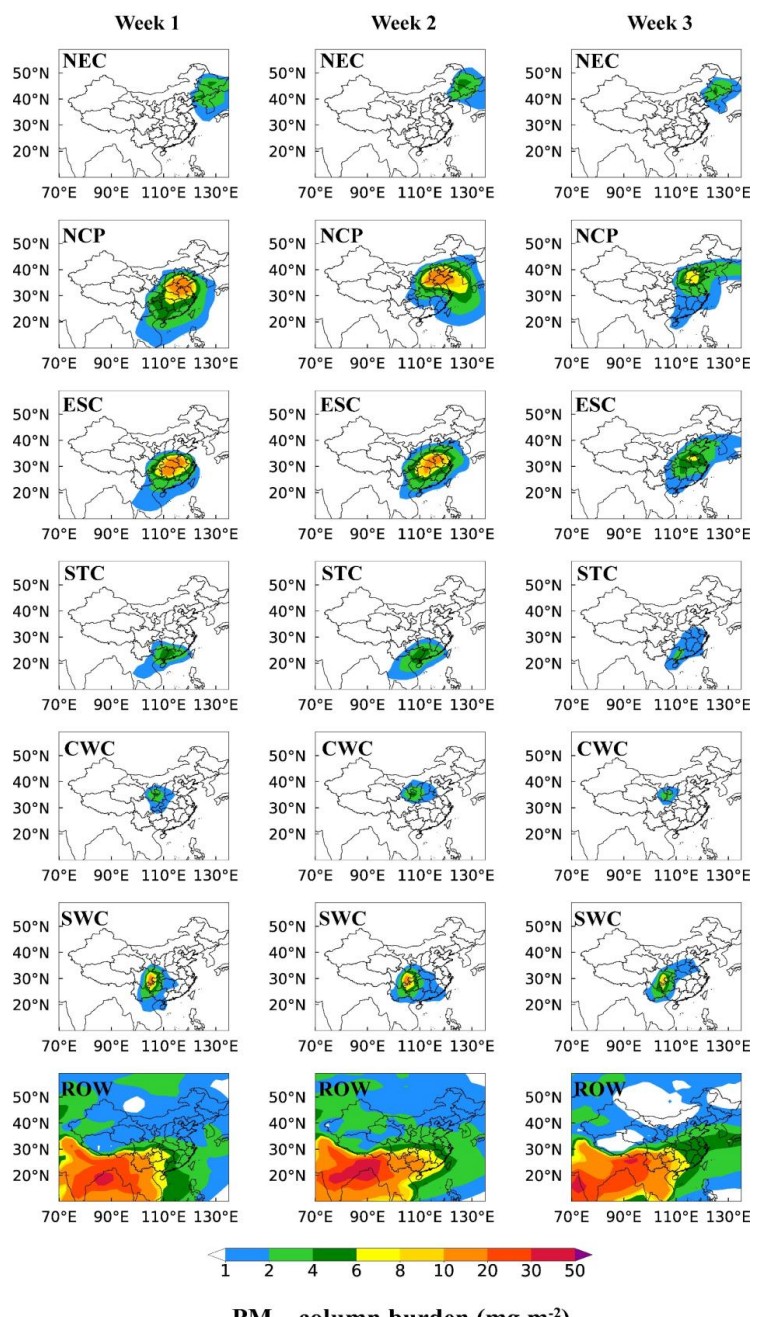

720

**Figure 4.** Spatial distribution of PM$_{2.5}$ column burden (mg m$^{-2}$) originating from the
six major source regions in China (NEC, NCP, ESC, STC, CWC and SWC) and sources
outside China (ROW) during the three time periods.

724



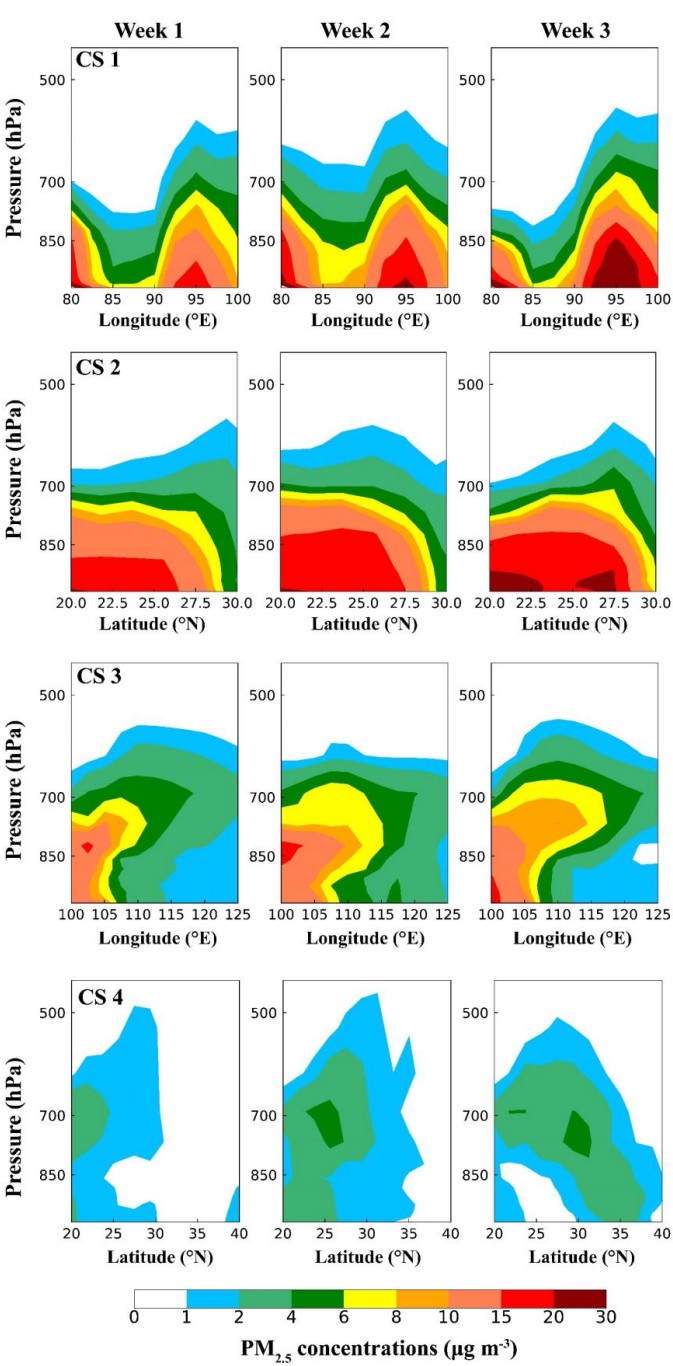

**Figure 5.** Vertical distributions of PM$_{2.5}$ concentrations (µg m$^{-3}$), originating from emissions outside China (i.e., ROW sources), across the latitudinal and/or longitudinal extents marked in Fig.1, respectively, during the three time periods.

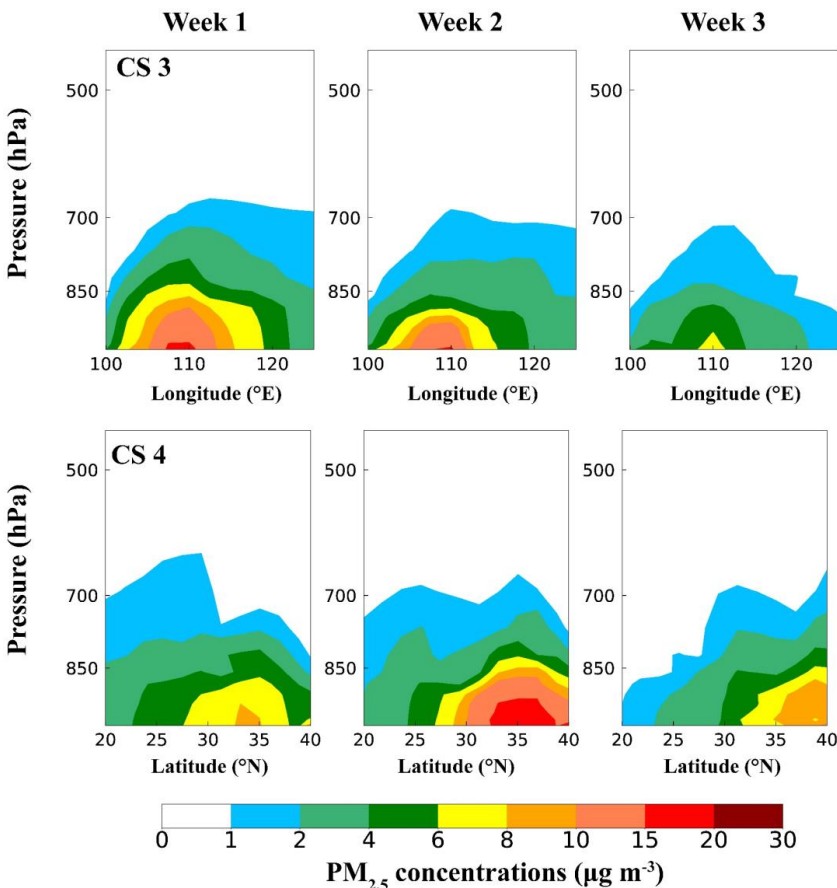

729

730

**Figure 6.** Vertical distributions of PM$_{2.5}$ concentrations (µg m$^{-3}$), originating from domestic emissions in China, across the latitudinal and/or longitudinal extents marked in Fig.1, respectively, during the three time periods. The values along CS 1 and CS 2 are negligibly small.

735

736



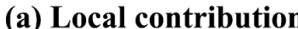

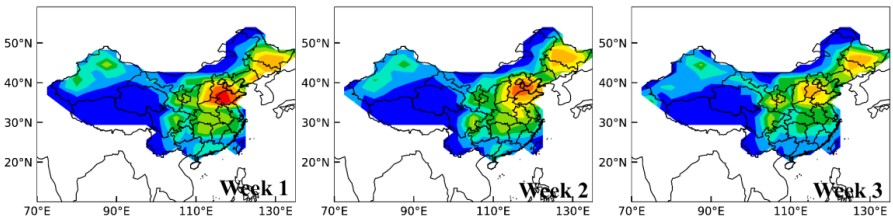

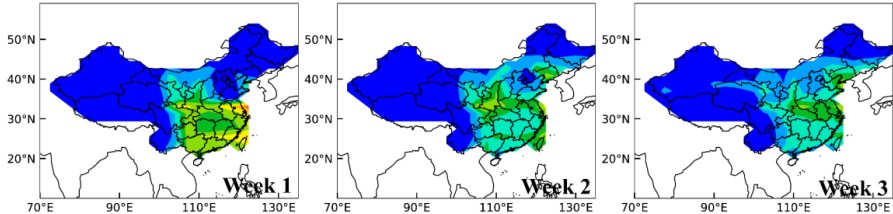

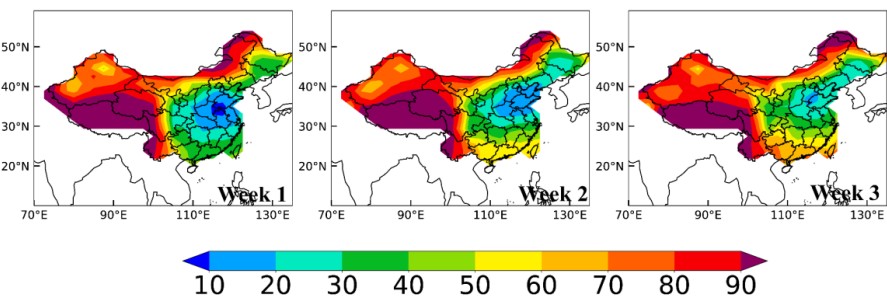

**Figure 7.** Relative contributions (%) of (a) local emissions, (b) the emissions from the
rest of China (RCN) and (c) all sources outside China (rest of the world, ROW) to $PM_{2.5}$
column burden during the three time periods.





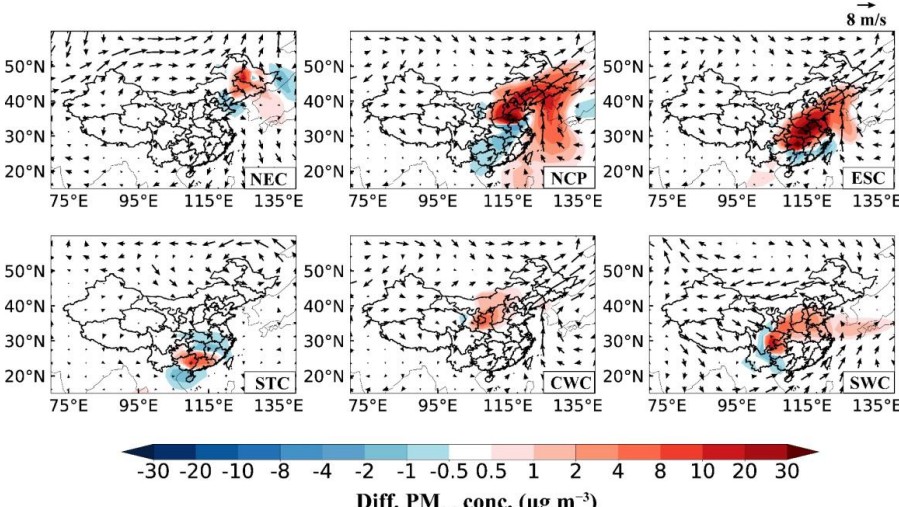

**Figure 8.** Composite differences in winds at 850 hPa (m s$^{-1}$) and near-surface PM$_{2.5}$ concentrations (µg m$^{-3}$) between polluted and normal days in February 2020. The polluted day is defined as the day with the highest daily PM$_{2.5}$ concentration in February 2020 in each receptor region in China.

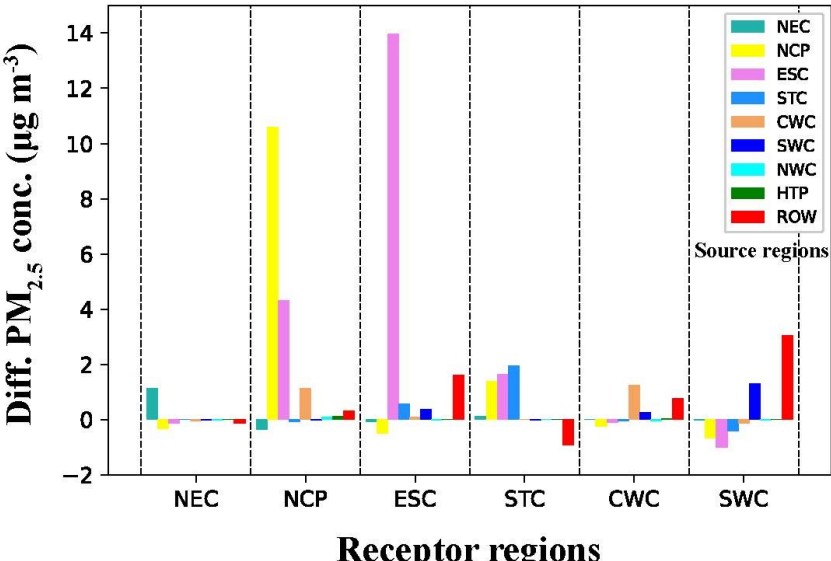

750

751

**Figure 9.** Composite differences in near-surface PM$_{2.5}$ concentrations (μg m$^{-3}$)
averaged over receptor regions (marked on the horizontal axis) in China between
polluted and normal days in February 2020 originating from individual source regions
(corresponding color bars in each column).