# Peer review of "Aerosol transport pathways and source attribution in China during the COVID-19 outbreak"

_Atmospheric Chemistry and Physics, 2021_

## Author Comment (AC1)

**Manuscript # acp-2021-328**

**Responses to Referee #1**

Ren et al. assess the contributions from local emissions and transport to $PM_{2.5}$ concentrations in Chinese regions during three periods when COVID-19 affected the socioeconomic activity of the country at the beginning of 2020. In principle, the topic is interesting and relevant, but I have major reservations concerning the chosen methods and the documentation thereof as well as the interpretation and discussion of the results.

We thank the editor for all the insightful comments. Below, please see our point-by-point response (in blue) to the specific comments and suggestions and the changes that have been made to the manuscript, in effort to take into account all the comments raised here.

The finding of regionally increasing $PM_{2.5}$ during the COVID-19 period is in light of lockdowns counterintuitive and needs a clearer discussion in the text. The authors state that it is due to transport from outside of China, but the quantified 4-10% of $PM_{2.5}$ transport into a Chinese region for polluted days and even the largest 40-66% regional contributions from transport during the lockdown, when local emission should be small, are no particularly convincing evidences, especially in light of the known poor model performance for $PM_{2.5}$ indicated by the authors.

Response:

Thanks for the suggestion. The polluted days are selected for "each receptor region" in China. Therefore, the large contribution from transboundary transport is only for some specific regions in China, e.g., Southwestern China. Also, the significant impacts from South and Southeast Asian emissions have been revealed in many previous studies.

We have now revised the sentences: "In this study, the most polluted day is defined as the day with the highest daily $PM_{2.5}$ concentration in February 2020 for each receptor region in China. The transport from outside of China only has a great impact on some specific regions in China. In Southwestern China, the relative contribution from ROW emissions, especially those from South and Southeast Asia, to the increment of $PM_{2.5}$ concentration during the most polluted days compared with normal days is more than 50%. It is consistent with the previous studies that emissions from South and Southeast Asia have an important impact on air quality in southwest China (Yang et al., 2017a; Zhu et al., 2016, 2017). For other receptor regions in China (Northeastern China, North China Plain, Eastern China, Southern China and Central-West China), $PM_{2.5}$ concentrations are largely contributed by local

emissions during the most polluted days compared with normal days. In the future with emissions reductions for better air quality in China, decreasing air pollution should consider aerosols from both Chinese local emissions and pollutant transport from outside of China."

The results need a more compelling interpretation, making better use of the knowledge of the impact of the meteorological conditions on $PM_{2.5}$, e.g., through a discussion in light of other studies. It would be useful to have a discussion section separate from the conclusions. This would allow to fully appreciate the limits and advances of this work compared to previous studies, and draw a clear and concise conclusion from this work.

Response:
    Thanks for the suggestion. We have now included such context in the discussion section as follows: "Many studies have examined the importance of meteorology on regional air quality during the COVID-19 lockdown period and emphasized that, when meteorology is unfavorable, abrupt emissions reductions cannot avoid severe air pollutions (Le et al. 2020; Sulaymon et al. 2021; Shen et al. 2021). Through model simulations, Le et al. (2020) found that abnormally high humidity promotes the heterogeneous chemistry of aerosols, which have contributed to the increase of $PM_{2.5}$ by 12% in northern China during the city lockdown period. Sulaymon et al. (2021) found that significant increase in $PM_{2.5}$ concentrations caused by unfavorable meteorological conditions in Beijing-Tianjin-Hebei region during the lockdown period based on Community Multiscale Air Quality (CMAQ) model simulations. By analyzing the observational data and model simulations, Shen et al. (2021) reported that 50% of the pollution episodes during the COVID-19 lockdown in Hubei of China were due to the stagnant meteorological conditions. Huang et al. (2020) found that the stagnant air conditions and enhanced atmospheric oxidizing capacity caused a severe haze event during the same time period. In line with previous studies, we also revealed the stagnant air condition under the anomalous high pressure system in the most polluted day over the North China Plain. In addition to the meteorological conditions, the effect of foreign transport was also raised in this study causing aerosol pollution in southwestern China during COVID-19 outbreak."

Specific comments

L. 51: "in December 2019" - give the time period of the outbreak in China
Response:
    As the epidemic broke out one after another in different areas, the outbreak time is a continuous time. We have now revised the text as follows: "The coronavirus disease 2019 (COVID-19) has spread worldwide since December 2019 and resulted in more than one million cases within the first four months

(Sharma et al., 2020; Dong et al., 2020)."

L. 53-54: I recommend removing "was the first country" from the sentence since it is not relevant for the scientific content, but say instead when the measures began and ended since this is indeed relevant for the interpretation of your findings.
Response:
   We have now revised the sentence to reflect this: "In order to curb the virus spread among humans, measures were taken by the Chinese government on January 23, 2020 to minimize the interaction among people, including strict isolation, prohibition of large-scale private and public gatherings, restriction of private and public transportation and even lockdown of cities (Tian et al., 2020; Wang et al., 2020)."

L. 62-63: revise sentence for clarity
Response:
   We have now revised the sentence: "The estimated NOx emission in eastern China was reduced by 60-70%, of which 70-80% was related to the reduced road traffic and 20-25% was from industrial enterprises shutdown during the COVID-19 lockdown period. However, severe air pollution events still occurred in East China during the COVID-19 lockdown, even though the anthropogenic emissions were greatly reduced (Huang et al., 2020)."

L. 66: "change" -> changes
Response:
   Revised.

L. 80-83: when did the haze occur? Does your simulation reproduce this event?
Response:
   In the study of Huang et al. (2020), the severe air pollution events occurred on February 11, 2020. Our model reproduced the pollution event at the same time and have now included such context in the discussion as follows: "When the $PM_{2.5}$ pollution occurred in the North China Plain on February 11, 2020, which was also reported as the polluted day in observations (Huang et al., 2020), the concentration of $PM_{2.5}$ was 16.1 μg m$^{-3}$ higher than that in normal days."

L. 114-116: If there are studies already, what does your work add to the previous knowledge?
Response:
   In the original text, "few" studies have focused on aerosol transport pathways and source attribution in China during the COVID-19 pandemic. Since the studies about the air quality during COVID-19 are increasing, we have emphasized our study that "Our study provides source apportionment of

aerosols covering the whole China and quantifies the contribution from foreign transport for the first time in the case of COVID-19 emission reductions."

L. 146: How many simulations did you perform over what time period?
Response:
    By adding an additional simulation in the revised manuscript, we now have two simulations with aerosol tagging but different emission assumptions: "The anthropogenic emissions used in the baseline simulation are derived from the MEIC (Multi-resolution Emission Inventory of China) inventory (Zheng et al., 2018), referred to here as the Baseline experiment. While emissions for the other countries use the SSP (Shared Socioeconomic Pathways) 2–4.5 scenario data set under CMIP6 (the Coupled Model Intercomparison Project Phase 6). Emissions in year 2017 are used as the baseline during the simulation period considering the time limit of MEIC inventory. To better estimate the impact of restricted human activities on emission reductions owing to COVID-19 lockdown (referred to as Covid experiment), we updated China's emission inventory from January to March 2020 based on the provincial total emission reduction ratio in Huang et al. (2020). Emissions from the transportation sector are decreased by 70%. The remaining emission reduction, by excluding transport reduction from the total emission reduction, are evenly distributed to other sectors, including industry, power plant, residential, international shipping and waste treatment from January to March 2020 compared to the baseline emission in 2017. Unless otherwise specified,all the results in this study are derived from the Covid experiment."

L. 150-151: There should be an argument why emissions from SSP2-4.5 are used here, even though more recent global emission data has been created (e.g., Lamboll et al., 2020)
Response:
    When we conducted the experiments, the latest global emission data has not been published. Applying the emissions from SSP2-4.5 can better compare with the simulations of CMIP6, which has been widely used in many previous studies (Lund et al. 2019; Lyakaremye et al. 2021).

L. 157-160: How were these emission estimates created? Please illustrate the results for the emissions and compare them to other new emission data. What is meant by „remaining reductions"?
Response:
    The emission reductions due to COVID-19 lockdown were updated based on dynamic economic and industrial activity levels, which has been applied in the previous studies (Huang et al., 2020). Emissions from the transportation sector are decreased by 70%. The remaining emission reduction, by excluding transport reduction from the total emission reduction, are evenly distributed to other sectors, including industry, power plant, residential, international shipping

and waste treatment from January to March 2020 compared to the baseline emission in 2017.

L. 166: "from April 2019 to March 2020 repeatedly for six years" this needs more words to explain what you did. How did you do for instance the initialisation? What is meant by repreating the simulation for six years?
Response:
    The simulations are integrated for 6 years with the first five years treated as model spin-up and the last year was analyzed.

L. 169: It would be more relevant to say which weeks had the most severe lockdowns and use this information to interpret the results.
Response:
    The lockdown was first implemented on January 23 in Wuhan, China. Subsequently, other regions in China took measures, and the lockdown of the whole country lasted for at least three weeks varying in different regions.

L. 191: What motivates the choice of these regions?
Response:
    The eight source areas are divided mainly according to the geographical location and subdivided on the basis of previous studies (Yang et al., 2017a).

L. 198-201: Were these nudged simulations to MERRA-2 as well? Then say so. Otherwise, it would be useful to say a few words on the performance of MERRA-2 over China as well.
Response:
    Yes. Many studies have nudged the model wind fields toward the MERRA-2 reanalysis in China (Zhuang et al., 2018; Yang et al., 2017a).

L. 212: I appreciate and encourage the open communication of uncertainties in modeling. A 50% underestimation of $PM_{2.5}$ is large. Given your focus on $PM_{2.5}$ in this study, how can you nevertheless trust the simulation, especially taking into account that nitrate and ammonium are known to be poorly represented in the same model (L. 217)? You revisit this point in the last paragraph of the conclusions, but I also missed guidance for the concrete implication of it there.
Response:
    We have now added the sentence to reflect this: "In majority of the climate models, the simulation of nitrate and ammonium aerosols are not included in the aerosol schemes, partly due to the complexity of calculation efficiency. For example, in many of the CMIP6 models, only two of them provide nitrate and ammonium mass mixing ratios. Many previous studies have evaluated the global climate models performance in reproducing aerosol concentrations (e.g., Fan et al., 2018; Shindell et al., 2013; Yang et al., 2017a,b). In general, the models can well simulate aerosols in North America and Europe but significantly underestimates aerosols in East Asia by about −36 to −58 %

compared with observations. It can lead to an underestimation of aerosols contributed by Chinese local emissions in magnitudes, but might not change the main conclusions of this study."

L. 227: State here the three time periods and motivate this choice.
Response:
    We now have added a description as follows. "Figure 3b presents spatial distributions of simulated mean column burden of $PM_{2.5}$ during the three time periods ('Week 1': January 30–February 5, 'Week 2': February 6–February 12 and 'Week 3': February 13–February 19), which had the largest number of newly-diagnosed COVID-19 cases."

L. 230 - 236: Say relative to what you make the comparisons.
Response:
    We have now revised the sentences: "Comparing to Week 3, Week 1 and Week 2 have higher $PM_{2.5}$ loading, with values in the range of 20–40 and 20–30 mg m$^{-2}$ in the North China Plain, Eastern China, and Southern China, while the $PM_{2.5}$ loading in Week 3 is relative lower than Week 1 and Week 2 with values ranging mostly from 10 to 20 mg m$^{-2}$."

L. 368: It would be helpful to state the date in the text, here and/or earlier.
Response:
    We now have added a description as follows. "When the $PM_{2.5}$ pollution occurred in the North China Plain on February 11, 2020, which was also reported as the polluted day in observations (Huang et al., 2020), the concentration of $PM_{2.5}$ was 16.1 μg m$^{-3}$ higher than that in normal days."

L. 273-374: 4-10% transport from outside of China on the most polluted day means that local emissions dominate. Maybe explicitly add the implication of your findings.
Response:
    Thanks for the suggestion. We have now included such discussions as follows: "The transport from outside of China only has a great impact on some specific regions in China. In Southwestern China, the relative contribution from ROW emissions, especially those from South and Southeast Asia, to the increment of $PM_{2.5}$ concentration during the most polluted days compared with normal days is more than 50%. It is consistent with the previous studies that emissions from South and Southeast Asia have an important impact on air quality in southwest China (Yang et al., 2017a; Zhu et al., 2016, 2017). For other receptor regions in China (Northeastern China, North China Plain, Eastern China, Southern China and Central-West China), $PM_{2.5}$ concentrations are largely contributed by local emissions during the most polluted days compared with normal days. In the future with emissions reductions for better air quality in China, decreasing air pollution should consider aerosols from both

Chinese local emissions and pollutant transport from outside of China."

Arrange the order of all figures following the order of references to them in the text.
Response:
    Thank you for your reminding, we have reorganized the order of figures.

Figure 1: What time period is meant here?
Response:
    The time period here refers to the three weeks of the study from January 30 to February 19, which had the largest number of newly-diagnosed COVID-19 cases.

Figure 2: What do the colors mean?
Response:
    The color is to distinguish the different weeks. We have now moved this figure to the supplement.

Table 1: State the dates of the weeks.
Response:
    We have now revised the sentences: "Table 1. Fractional contributions of emissions from nine tagged source regions (vertical axis) to mean $PM_{2.5}$ column burden in eight receptor regions (horizontal axis) during the three time periods ('Week 1': January 30–February 5, 'Week 2': February 6–February 12 and 'Week 3': February 13–February 19)."

Reference:

[revised manuscript text omitted]

---

## Author Comment (AC2)

**Responses to Referee #2**

The authors investigated aerosol transport pathways in China during COVID-19. They established the source-receptor relationships among various regions of China using the CAM5 model with the capability of aerosol source tagging. The model system was developed by the same group of this paper and was evaluated in their previous studies. This work suggests that local emissions contribute largely to the severe aerosol pollution in North China Plain and Eastern China during COVID along with moderate impacts from unfavorable meteorological conditions. Overall, this paper reads well and provides interesting results, which could benefit the design of air pollution regulation strategies in China. I have two major concerns about the manuscript in its current form, which need to be resolved before it can be accepted for publication.

We thank the editor for all the insightful comments. Below, please see our point-by-point response (in blue) to the specific comments and suggestions and the changes that have been made to the manuscript, in effort to take into account all the comments raised here.

The first problem is that the CAM5 model used in this work cannot simulate nitrate and ammonium aerosols, while these compositions account for a large proportion of aerosols over China currently. Please provide detailed explanations and discussions on how this model deficiency could impact the main conclusions of this work.

Response:
    Thanks for the suggestion. We have now added the following sentences in the discussion section: "In majority of the climate models, the simulation of nitrate and ammonium aerosols are not included in the aerosol schemes, partly due to the complexity of calculation efficiency. For example, in many of the CMIP6 models, only two of them provide nitrate and ammonium mass mixing ratios. Many previous studies have evaluated the global climate models in reproducing aerosol concentrations (e.g., Fan et al., 2018; Shindell et al., 2013; Yang et al., 2017a, b). In general, the models can well simulate aerosols in North America and Europe but significantly underestimates aerosols in East Asia by about $-36$ to $-58\,\%$ compared with observations. It can lead to an underestimation of aerosols contributed by Chinese local emissions in magnitudes, but might not change the main conclusions of this study."

The second problem is that the focus of this work is the aerosol source

attribution during COVID. However, the authors did not discuss much the special findings in this special period. Compared to previous literature, are there any novel results and conclusions of the contributions from local/nonlocal sources to aerosol pollutions during this period with low emission levels? And what's the implication for air pollution control policies in China, especially considering that the anthropogenic emissions will be rapidly reduced in the future?

Response:

Thanks for the suggestion. We have now included such context in the discussion section as follows: "Source tagging and apportionment is an effective way to establish aerosol source-receptor relationships, which is conducive to both scientific research and emission control strategies (Yu et al., 2012). Previous studies only focused on regional transport of aerosols, very few studies have explored the aerosol transport pathways and source attribution covering the whole China during the COVID-19 pandemic. The COVID-19 pandemic disrupted human activities and lead to abrupt reductions in anthropogenic emissions. This study first investigated the source contributions to $PM_{2.5}$ over various regions covering the whole China during the COVID-19 pandemic. We pay attention not only to local emissions, but also to the impacts from regional and foreign transport of aerosols."

In the revised manuscript, we added an additional experiment to better reflect variations of contributions from local/nonlocal sources to aerosol pollutions during this period with low emission levels. "The anthropogenic emissions used in the baseline simulation are derived from the MEIC (Multi-resolution Emission Inventory of China) inventory (Zheng et al., 2018), referred to here as the Baseline experiment. While emissions for the other countries use the SSP (Shared Socioeconomic Pathways) 2–4.5 scenario data set under CMIP6 (the Coupled Model Intercomparison Project Phase 6). Emissions in year 2017 are used as the baseline during the simulation period considering the time limit of MEIC inventory."

"To highlight the roles of regional and foreign transport, the differences between Covid and Baseline simulations in relative contributions to $PM_{2.5}$ burden from local, region (RCN) and foreign (ROW) emissions are given in Figure S1. During the COVID-19 period, the local and RCN emission contributions to $PM_{2.5}$ were 1–4% lower than that in Base experiment over NCP and NEC. In Eastern China, the contribution from the local emissions decreased by 3–4% compared with Base experiment, while the contribution from ROW increased by more than 5%. In Southern China, 50–70% of the $PM_{2.5}$ burden is contributed by emissions from ROW in Base experiment. During the COVID-19 period with low emission levels, the contribution from ROW to $PM_{2.5}$ burden in Southern China had an increase of more than 5%. It indicates that the important role of transboundary transport needs to be considered when controlling local emissions to improve air quality in the near future."

[Figure]

Figure S1. Relative contributions (%) in Baseline simulation (left) and differences in relative contributions (%) between Covid and Baseline simulations (right) of local emissions (top), the emissions from the rest of China (RCN) (middle) and all sources outside China (rest of the world, ROW) (bottom) to PM$_{2.5}$ column burden in February 2020.

Reference:

Fan, T., Liu, X., Ma, P.-L., Zhang, Q., Li, Z., Jiang, Y., Zhang, F., Zhao, C., Yang, X., Wu, F., and Wang, Y.: Emission or atmospheric processes? An attempt to attribute the source of large bias of aerosols in eastern China simulated by global climate models, Atmos. Chem. Phys., 18, 1395–1417, https://doi.org/10.5194/acp-18-1395-2018, 2018.

Shindell, D. T., Lamarque, J.-F., Schulz, M., Flanner, M., Jiao, C., Chin, M., Young, P. J., Lee, Y. H., Rotstayn, L., Mahowald, N., Milly, G., Faluvegi, G., Balkanski, Y., Collins, W. J., Conley, A. J., Dalsoren, S., Easter, R., Ghan, S., Horowitz, L., Liu, X., Myhre, G., Nagashima, T., Naik, V., Rumbold, S. T., Skeie, R., Sudo, K., Szopa, S., Takemura, T., Voulgarakis, A., Yoon, J.-H., and Lo, F.: Radiative forcing in the ACCMIP historical and future climate simulations, Atmos. Chem. Phys., 13, 2939–2974, https://doi.org/10.5194/acp-13-2939-2013, 2013.

Yang, Y., Wang, H., Smith, S. J., Ma, P. L., Rasch, P. J.: Source attribution of

black carbon and its direct radiative forcing in China, Atmospheric Chemistry and Physics, 17, 4319–4336, https://doi.org/10.5194/acp-17-4319-2017, 2017a.

Yang, Y., Wang, H., Smith, S. J., Easter, R., Ma, P. L., Qian, Y., Yu, H., Li, C., Rasch, P. J.: Global source attribution of sulfate concentration and direct and indirect radiative forcing, Atmospheric Chemistry and Physics, 17, 8903–8922, https://doi.org/10.5194/acp-17-8903-2017, 2017b.

Yu, H. B., Remer, L. A., Chin, M., Bian, H. S., Tan, Q., Yuan, T. L., Zhang, Y.: Aerosols from overseas rival domestic emissions over North America, Science, 337, 566–569, https://doi.org/10.1126/science.1217576, 2012.

Zheng, B., Tong, D., Li, M., Liu, F., Hong, C., Geng, G., Li, H., Li, X., Peng, L., Qi, J., Yan, L., Zhang, Y., Zhao, H., Zheng, Y., He, K., and Zhang, Q.: Trends in China's anthropogenic emissions since 2010 as the consequence of clean air actions, Atmospheric Chemistry and Physics, 18, 14095–14111, https://doi.org/10.5194/acp-18-14095-2018, 2018.